# Our Experience with Hydrogel-Coated Trisacryl Microspheres in Uterine Artery Embolization for the Treatment of Symptomatic Uterine Fibroids and Adenomyosis: A Follow-Up of 11 Years

**DOI:** 10.3390/jpm13091385

**Published:** 2023-09-15

**Authors:** Panagiotis Tsikouras, Efthymios Oikonomou, George Tsatsaris, Anastasia Bothou, Dimitrios Kyriakou, Konstantinos Nikolettos, Theopi Nalmbanti, Panagiotis Peitsidis, Grigorios Trypsanis, George Iatrakis, Nikolaos Nikolettos, Vasileios Souftas

**Affiliations:** 1Department of Obstetrics and Gynecology, Democritus University of Thrace, 68100 Alexandroupolis, Greece; eftoikonomou@outlook.com (E.O.); tsatsarisg3@gmail.com (G.T.); natashabothou@windowslive.com (A.B.); dimitriskyrkdk@gmail.com (D.K.); k.nikolettos@yahoo.gr (K.N.); theonalmpanti@hotmail.com (T.N.); peitsidiobgyn@gmail.com (P.P.); giatrakis@uniwa.gr (G.I.); nnikolet@med.duth.gr (N.N.); 2Department of Medical Statistics, Democritus University of Thrace, 68100 Alexandroupolis, Greece; gtryps@med.duth.gr; 3Department of Interventional Radiology, Democritus University of Thrace, 68100 Alexandroupolis, Greece; vsouftas@med.duth.gr

**Keywords:** fibroids, adenomyosis, uterine artery embolization, hydrogel embolization agents

## Abstract

Uterine artery embolization (UAE) for the treatment of symptomatic uterine fibroids and non-controllable adenomyosis symptoms is a relatively new procedure for organ-preserving therapy. These benign conditions can become symptomatic in about 30% of women between the ages of 35 and 50. The purpose of the UAE either for fibroids or adenomyosis is the elimination of blood loss, the reduction in pain, and bulky or rectal pressure symptoms. The purpose of this study is to present our experience in UAE with the use of hydrogel-coated tris acryl microspheres for the treatment of symptomatic uterine fibroids and adenomyosis.

## 1. Introduction

Before menopause, 20% of visits to a gynecologist are due to abnormal menstrual bleeding (AMB) from the genital tract. Abnormal perimenopausal uterine bleeding (AUB) can be the result of many pathological causes, and the primary goal of clinical evaluation and management of AUB is to diagnose the underlying cause and apply the most efficient and less invasive treatment [1,2]. The majority of fibroids (60–70%) are asymptomatic, and approximately 80% of women by the time of menopause have radiologic or pathologic evidence of fibroids. However, only 25% of women with fibroids have severe symptoms and need therapy [3,4]. Although hysterectomy offers a definitive solution, women may prefer to preserve their uterus due to various reasons. Despite the fact that the removal of fibroids, either abdominally or laparoscopically, is still the most recommended surgical procedure, conservative therapies, including several medical treatment options, are preferred by the majority of women. Medical therapies are based on biochemical interactions that influence ovarian steroids and fibroid growth. Thus, the goal of medical therapies is the manipulation of the hormonal environment to achieve fibroid shrinkage and bleeding regulation [5,6,7,8]. Furthermore, conservative nonsurgical procedures include uterine artery embolization, which shrinks fibroids. This is a relatively new procedure for organ-preserving therapy that could be implemented in women with symptomatic uterine fibroids, including abnormal vaginal bleeding (most common), frequent urination, rectal pressure, and pelvic pain due to their anatomic location, size, number, and degenerative changes. The same method could be used in the treatment of symptomatic adenomyosis [5,6,7,8].

The referred frequency of fibroids, which may become symptomatic, is reported in approximately 30% of women between the ages of 35 and 50 years [7]. With the exception of Great Britain, France and above all the USA, the application of the procedure in European countries is struggling to gain acceptance. The reason for this is multifaceted [8]. On the one hand, in this process, the interventional radiologist treats a patient that is not normally encountered in their other everyday work [8,9,10]. More than with other interventions, the interventional radiologist is required to get involved in the determination of the indication, implementation and post-interventional care of the patients, since the clinically attending gynecologist is not required to have any extensive knowledge of the post-interventional course of such a new procedure as embolization. Uterine artery embolization (UAE) is a safe and effective method in treating symptomatic uterine fibroids, especially with regard to uterine bleeding and pelvic discomfort, which improve without the need for surgical procedures [10,11,12]. The key to the successful application of this method lies in the close collaboration between the interventional radiologist and gynecologist, before and after embolization. Although UAE is associated with improved public health, no prospective randomized trials have been investigated and performed to evaluate the safety and effectiveness of UAE [12,13,14]. At the University Hospital of Alexandroupolis, there was a perfect cooperation between gynecologists and radiologists, both in the indication for the application of transcatheter embolization of the uterine arteries and in the post-operative follow-up of women. In the present study, the clinical laboratory and imaging data obtained after the embolization of women in our hospital with specific acrylic beads (Embozene, CeloNova Biosciences, Newnan, GA, USA) are analyzed.

## 2. Material and Methods

Patients were treated between November 2013 and December 2017. From the patient history, the main clinical symptoms noted based on patient records included menometrorrhagia, dysmenorrhea, dyspareunia and symptoms attributed to bulky disease or pressure on pelvic organs. Four patients who had undergone fibromyectomy followed by leiomyoma recurrence were also included. In all cases, bilateral UAE was performed in one session, with percutaneous puncture of the right common femoral artery and insertion of a 4F catheter into the uterine arteries. The left uterine artery was initially selectively accessed with the crossover technique and, when the catheter bypassed the arteries supplying the vagina and cervix, administration of the embolizing particles was initiated. In the same manner, catheterization and embolization of the right uterine artery were then performed.

Post-menopausal females, patients with serious comorbidities, patients wishing to preserve their fertility, patients with a known allergy to the contrast agent utilized during the procedure and patients with a suspected malignant condition were excluded from the present study. Pedunculated subserosal fibroids were excluded due to the risk of detachment from uterus postinterventional. All patients provided written informed consent prior to the UAE. Ethical approval for this procedure was confirmed by the ethics committee of the University Hospital in Alexandroupolis, Democritus University of Thrace (Alexandroupolis, Greece; reference no. 8/37 10/10/13). UAE was performed by means of percutaneous catheterization from the femoral artery, applying the fluoroscopic approach to the uterine artery, and the injection of media such as PVA (polyvinyl alcohol) particles or hydrogels to coat this acrylic gelatin microspheres. UAE is a non-surgical procedure which is performed under sedation using a catheter which is passed through the uterine arteries, and embolic agents are targeted at the symptomatic fibroids. The procedure’s goal is to decrease blood flow, causing the infarction and eventual shrinking of the fibroids. Its effectiveness in reducing bleeding reaches 80–100%, and its effectiveness in reducing pressure symptoms reaches 40–60%. In all cases, Embozene acrylic beads were used, with a slow infusion flow through the lumen of the microcatheter. The size of the beads for the fibroids was 700 µm. In a few cases, larger beads (900 μm) were also used.

In adenomyosis, only Embozene 500 μm acrylic beads were used. Fibroids have wider arterioles than the myometrium, and the goal of the UAE was to induce ischemia in the fibroids, while causing the least possible ischemic injury to the myometrium. In adenomyosis lesions (focal or diffuse), the arterial supply is carried out with arterioles of the same size as those of the myometrium (in adenomyomas with a characteristic spiral morphology and size of 350–500 μm), and so with embolization, the myometrium is ischemic as a whole (diameter of arterioles 200–500 μm), while the adenomyosis lesions are ischemic to a greater extent [8,9,10,11,12,13,14]. When small plunger sizes are used, they risk causing uterine necrosis with potentially tragic consequences. The amounts of pellets used varied and depended on the size of the lesions (from four 2 mL vials to 30 2 mL vials).

The criterion for stopping the administration of the beads was identifying the beginning of the regurgitation of beads towards the proximal part of the uterine artery (during the infusion with a very slow flow), in which case the administration was interrupted for 10 min and “complete stasis” after the repetition of the infusion was established [8,9,10,11,12,13,14].

In adenomyosis, the final criterion was “near-complete stasis” after stopping for 10 min (i.e., continued flow very slowly with regression to minimal impulses. Pellet rebounds were not to reach the efferent of the arteries for the cervix and vagina, and certainly not more centrally. We used CELLONOVA acrylic spheres (precise calibrated hydrogel-coated acrylic spheres, with a sphere diameter of 700 μm, and in the very large fibroids additionally/subsequently with a diameter of 900 μm).

In the cases of pure adenomyosis (diffuse and/or focal), the pellets had a diameter of 500 μm, while in the mixed type I started with pellets of 700 μm and continued with pellets of 500 μm, up to the limit of almost complete stasis (in the endometrial part of the respective uterine artery).

To estimate inflammatory response, C-reactive protein (CRP), white blood count (WBC), temperature (TEMP) and pain score measurements were recorded one day prior to and two days following UAE in all participants. Discharge was allowed after inflammatory parameter reduction.

## 3. Statistical Analysis

Statistical analysis of the data was performed using IBM Statistical Package for Social Sciences (SPSS), version 19.0 (IBM Corp., Armonk, NY, USA). The normality of quantitative variables was tested with the Kolmogorov–Smirnov test. Normally distributed quantitative variables were expressed as mean ± standard deviation (SD), while non-normally distributed quantitative variables were expressed as median value and interquartile range (25th to 75th percentile). The differences between pre- and post-UAE values were examined using paired-sample *t*-tests. ANOVA for repeated measurements or the Friedman test was used to analyze changes within groups over time, while ANOVA or the Kruskal–Wallis test was used to analyze differences between groups. The association between two quantitative variables was investigated using Pearson’s r correlation coefficient. All tests were two-tailed, and *p*-values under 0.05 were considered statistically significant.

## 4. Results

A total of 270 premenopausal women aged 33–55 years (mean 47.02 ± 4.78) were enrolled in this study. The mean uterine dimensions before and after embolism were 13.5 cm and 4.3 cm, respectively.

### 4.1. Pre- and Post-UAE Uterus Dimension and Fibroid Volume

Two hundred and seventy-two women aged between 33 and 61 years, with a mean age of 47.02 ± 4.78 years, were enrolled in the study. The mean uterine dimensions before and after embolism were 13.57 ± 3.01 cm and 4.31 ± 1.98 cm, respectively, while the mean fibroid volume before and after embolism was 78.95 ± 25.75 cm^3^ and 37.57 ± 38.85 cm^3^, respectively. The comparison of the mean uterine dimensions and fibroid volume before and after the embolism revealed a statistically significant reduction in uterus dimensions by 67.76% (*p* < 0.001) and fibroid volume by 52.41% (*p* < 0.001) after UAE. The largest fibroid was 159.2 cm^3^ in volume, and after embolism it was 99.4 cm^3^. The percentage of reduction in the largest fibroid’s volume among these women was 37.56%. These results indicate that the method was successful and the fibroids were reduced in volume (Figure 1).

### 4.2. Inflammatory Parameters throughout the Follow-Up Time

Throughout the follow-up time, CRP, temperature and WBC values exhibited significant changes within all three examined parameters according to UAE with hydrogel particles. In particular, a statistically significant gradual elevation of all CRP, temperature and WBC values was initially observed, reaching their peak value on the third post-UAE day, and returning to the pre-UAE levels on the fifth post-UAE day. No statistically significant differences in CRP, temperature and WBC values, at any time point, were found between the three groups according to hydrogel particles (Table 1A–C and Figure 2A–C).

### 4.3. Inflammatory Parameters in Relation to Uterus Dimensions and Fibroid Volume

In the sequence, the association of pre-UAE values of uterus dimensions and fibroid volume with maximum values of CRP, temperature and WBC was established. No statistically significant correlation of uterus dimension and fibroid volume with maximum values of CRP, temperature and WBC was found, with the exception of the weak positive correlation of fibroid volume with the maximum value of WBC (r = 0.141, *p* = 0.020) (Table 2).

### 4.4. Inflammatory Parameters in Relation to the Size of the Embolization Agent

The association of the molecular particle size of the embolization agent with the maximum values of CRP, temperature and WBC was also established. Sixteen patients received an agent with a molecular size of 500–700 µm, 242 received an agent with a molecular size of 700 µm, and 14 patients received an agent with a molecular size of 700–900 µm. Τhere was no statistically significant association between molecular size and CRP values (*p* = 0.779), temperature (*p* = 0.941) and WBC (*p* = 0.680) (Table 3).

## 5. Discussion

The identification of abnormalities in menstrual function during the transition to menopause presents an ongoing challenge for both women and their healthcare providers. During perimenopause, which is commonly encountered by the majority of women at some stage, there can be instances of extended menstrual bleeding, frequent or infrequent menstruation, and irregular menstrual cycles.

Although the typical description of the menopausal transition as a stage first marked by increased variability in menstrual cycle lengths followed by enhanced frequency of very long cycles until permanent amenorrhea describes the experience of the majority of women, marked differences occur in the magnitude of change in women’s menstrual experience [14,15,16,17,18]. Roughly 15% to 25% of women encounter minimal or negligible alterations in their menstrual regularity prior to reaching their final menstrual period. The FIGO 2011 classification of abnormal uterine bleeding facilitates investigations related to the cause, evaluation, management and treatment of abnormal uterine bleeding [19]. Irrespective of the reason for bleeding, a hysterectomy remains the definitive therapy option [20,21,22,23,24]. For women who are affected by uterine fibroids and prefer conservative management, several medical treatment options are recommended that target fibroids by manipulating their hormonal environment. It is well established that fibroids are hormonally responsive based on the fact that fibroid growth is dependent on the ovarian steroid hormones. Local growth factors are involved in fibroid growth and may mediate the growth effects of steroid hormones on the uterus. Also, genetic factors like Reed’s Bannayan syndrome, Cowden syndrome, and hereditary leiomyomatosis, chromosome 12 translocations that involve chromosome 14, vascular abnormalities and response to injury may play roles in fibroid growth [20,21,22,23,24]. A hysterectomy is associated with the significant risks of every surgical procedure, which include infection, thromboembolic diseases, postoperative bleeding and also a high cost and postoperative morbidity. In some women, hysterectomy can have additional benefits, e.g., the recurrence of fibroids and the risks of hormone replacement therapy. Others see hysterectomy as a mutilating intervention impacting their femininity and sexuality, and seek out alternative solutions, even though the reduction in flatulence indicates improvements in quality of life after hysterectomy [25,26,27,28]. The desire for minimally invasive alternatives to hysterectomy prompted the preference for the removal of uterine fibroids from the uterus and surrounding structures. This was the first surgical approach to conserve the uterus based on fibroid location. Although this procedure is effective concerning the reduction in menorrhagia and pelvic pressure, it is associated with a risk of recurrence. According to the literature, it was observed that one-third of patients who underwent abdominal fibromectomy required a second surgical procedure within a decade.

Further risks for future procedures include small uterus size and weight gain demonstrate a correlation with peripheral estrogen exposure. The removal of fibroids is associated with many key risks for pregnancy, including endometrial adhesion anemia, a risk of hysterectomy and increased risk of uterine rupture (0.002%), which is low compared to a previous classical cesarean section, at 3.7% in a subsequent pregnancy. Fibroids may lead to pregnancy complications and are also associated with infertility. Fibroid presence is the only pathological finding in less than 3% of women with infertility problems, and approximately 11% of pregnant women have fibroids [28,29,30,31,32].

The experience of pregnancies after UAE for obstetric bleeding is positive, but this finding is not extended to cases of UAE for fibroids. While there are several studies with good results, there are case reports with complications. UAE is associated with placental abnormalities. Live births occurred in 41% of pregnancies and approximately 21% ongoing beyond 30 weeks. In addition, there is also the possibility (2–7%) of embolism and the destruction or perfusion of the ovaries, resulting in premature ovarian functional failure [28,29,30,31,32]. Results are expressed as clinical data (improvement of bleeding and pressure symptoms) and as imaging (decreasing the size of the uterus and fibroids). Sometimes, even a slight reduction in the size of the fibroids can make a significant difference in symptoms. The success rate after the intervention is achieved in 84–100% of cases. Reintervention rates after UAE have been documented and reported as 9% at 1 year and 28% at 5 years according to the published literature versus surgical procedures [28,29,30,31,32]. According to the international literature, during UAE, particulate materials, including PVA polyvinyl alcohol particles, gelatin-coated trisacryl polymer microspheres and hydrogel polymer microspheres, can be delivered. In our study, participants used were administered only hydrogel polymer microspheres to produce ischemic damage in fibroids without causing permanent damage to the uterus, which could lead to an emergency hysterectomy. A bilateral femoral approach is preferred to reduce radiation exposure and procedural time. Identification using subtraction angiography can be performed to confirm that no vascular abnormalities are present. Technical difficulties may occur as a result of an anatomical variation arterial spasm or the current use of a gonadotropin-releasing hormone agonist. Failure may sometimes occur due to uterine perfusion from the collateral ovarian vasculature. Uterine cramping may be reduced via the administration of nonsteroid anti-inflammatory drugs.

The main technical problems are the difficulties in catheterization and spasming of the uterine artery. The reduction in the size of fibroids depends on the degree of their degeneration. Fibroids with good vascularity, as seen on MRI, are expected to shrink more than those that are degenerated. In general, after UAE, fibroids shrink by 40–70% and the size of the uterus shrinks by 40–60%. Over time, the shrinkage is increased. Recession of symptoms is expected at a rate approaching 90%. Specifically, menorrhagia subsides in 81–100% of cases, while pressure symptoms (flatulence, weight in the pelvis and frequency) decrease at percentages of 61–90%. Complications related to uterine embolization in the treatment of fibroids depend on how the outcome is calculated. The size of the uterus does not seem to be a decisive factor, because a remission of symptoms is observed just as often in patients with a uterus larger than 24 cm. These results are also confirmed by studies of the last two years, which state that patients are satisfied at rates of 85–90% [26,27,28,29,30,31,32].

According to the literature, there is a 40–70% volume reduction in fibroids within 3 and 6 months, respectively, and a 72–97% improvement in symptoms. Based on reports from the international literature, most complications occur in cases where the uterus is small and has an upper uterine size and in the uteruses whose size is massively enlarged. In large uteruses, more extensive collateral circulation exists, which may be accompanied by additional extensive infarction of tissue, possibly leading to increased necrosis-related complications [26,27,28]. The percentage of women who will need a hysterectomy is 4–30%, especially in cases with less than 10% shrinkage postintervention. There is no association between pretreatment uterine volume and clinical course. Postprocedural treatment failure was reported in cases with previous pelvic surgery [26,27,28,29,30,31,32]. This is all the more important since the benign nature of the underlying condition means that complications can only be tolerated within very narrow limits. The most serious complication is definitely the secondary infection of the myoma or the uterus, possibly with the development of sepsis [32,33,34]. We analyzed the development of these values in 270 patients with a normal course in order to describe the course of these parameters after embolization treatment, which can be associated with quantitatively not insignificant tissue necrosis, even under normal conditions. The time of observation was routinely set when the inflammatory parameters decreased.

We therefore routinely record the development of general inflammatory parameters in our patients during their clinical stay associated with embolization agents. Spherical embolization particles are devices that are left behind in the human body; it is essential that they do not cause any unnecessary foreign body reaction, inflammation, or local vessel irritation at the points of contact with the body. This is especially important when treating diseases in which additional postprocedural local vessel inflammation is harmful.

Embozene™ microspheres consist of a hydrogel core coated with CeloNova BioSciences’ proprietary polymer, Polyzene^®^-F, and offer biocompatibility, i.e., close to bioneutrality because they do not cause inflammatory reactions, do not allow platelet adhesion and are resistant to bacteria. Embozene™ microspheres are the only spherical embolic platform with a size of 250–900 µm which can pass through a microcatheter. Embozene™ microspheres not only retain their original volume but also immediately regain their original volume upon exit without any deformation or fragmentation. The frequency of UAE complications is very low. In general, the complications concern either the catheterization or the effects of the ischemia of the uterus that can cause necrosis of the fibroids with the appearance of a septic picture and their expulsion [8,28,29,30,31,32,33,34]. Finally, embolization of other organs, especially the ovaries, can also occur. Deaths reported after embolization are extremely rare (1:1600), and mainly involve pulmonary embolism, which can occur due to the effect of necrotic tissues on the activation of the coagulation mechanism, and upon infection. Catheterization complications are rare (<1%), such as hematoma formation, contrast allergies, and pseudoaneurysms or separation in the vessel. Although pregnancies after UAE have been reported, the desire to have children is a contraindication, because there is a risk of ovarian failure and the effects on fertility and pregnancy are unknown.

Looking at the literature, the following questions arise

Is there a correlation between the molecular size or quantity of the embolization agent and inflammatory factors?Does a correlation exist between the fibroid volume and inflammatory factors [6,28,29,30,31,32,33,34]?

According to our findings, it was found that the volume of fibroids significantly affects the CRP value on the second post interventional day, whilst also impacting the size reduction rate of fibroids, especially the largest ones, after embolization.

In Figure 1, no association between uterus dimensions and fibroid volume and the maximum values of CRP, TEMP and WBC is observed. However, we confirm a positive correlation of fibroid volume with the maximum value of WBC. In Table 2, no association between inflammatory factors and the amount of contrast agent and no relation with the molecular size of the embolization agent (hydrogel particles) is confirmed in our participants (Table 1A–C). The explanation for this is that the administered embolization agent substance is an isotone solution and leads to no disadvantages in the vessel intima [8,34,35,36,37,38].

However, these findings are combined with a study limitation concerning the relationship between molecular particles size and the quantity of the substance because the sample is inhomogeneous. In total, 240 participants received an embolization agent with a molecular size of 700 μm, 16 received an embolization agent with a molecular size of at least 500–700, and 14 received an embolization agent with a molecular size of 700–900 μm (Table 3).

Major complications of postprocedural UAE including chronic vaginal discharge due to fluid accumulation from infarcted fibroids, fibroid extrusion, ovarian failure, high fever and nausea necessitating hospitalization, vomiting, malaise, anorexia, sepsis, and death are estimated in 1–5% of cases, while in 1%, hysterectomy is necessary in cases in which severe postembolization syndrome occurs, especially with a large amount of devascularized tissue.

In all of our study participants, no side effects were identified, and the life quality satisfaction level checked via questionnaires and clinical and laboratory examinations was high.

From our results, it can be concluded that in the case of uncomplicated fibroid embolization, a significant increase in inflammatory parameters can be considered normal as an expression of fibroid necrosis. In numerous embolization treatments worldwide, as an experimental procedure, post-interventional pain needs to be regarded as uncritical. UAE has been proven to be associated with an equivalent reduction in fibroid symptoms, leading to improvements in quality of life, and is associated with a lower risk of infection and complications than hysterectomy. More future metacentric studies are necessary to investigate these scientific points in more homogenous participant samples to define guidelines and clinical protocols.

## Figures and Tables

**Figure 1 jpm-13-01385-f001:**
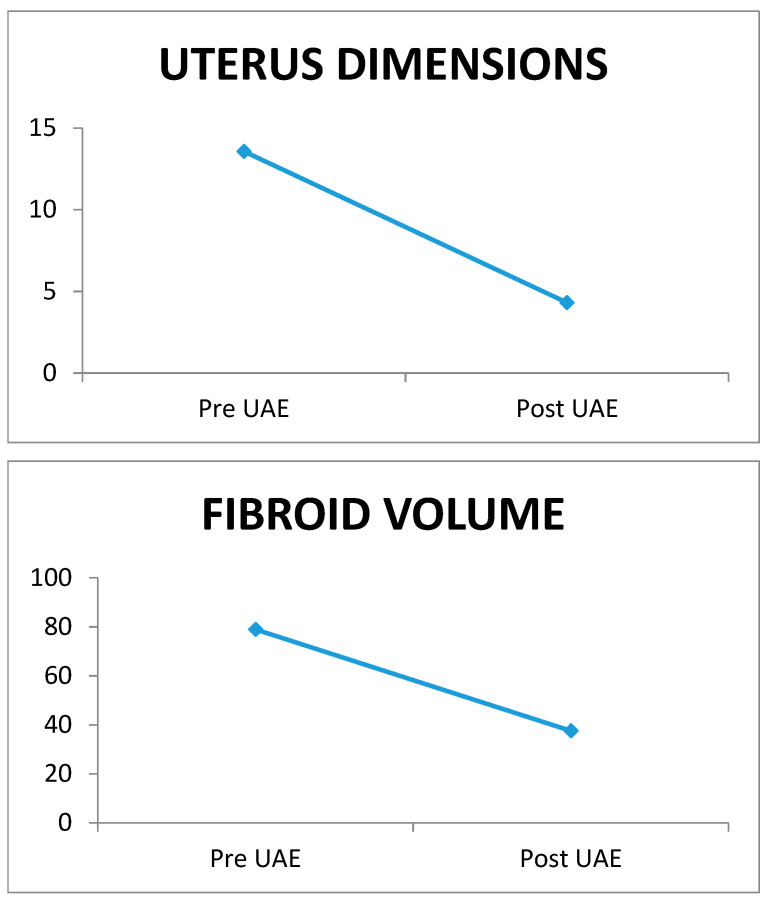
Uterus dimensions and fibroid volume pre- and post-UAE.

**Figure 2 jpm-13-01385-f002:**
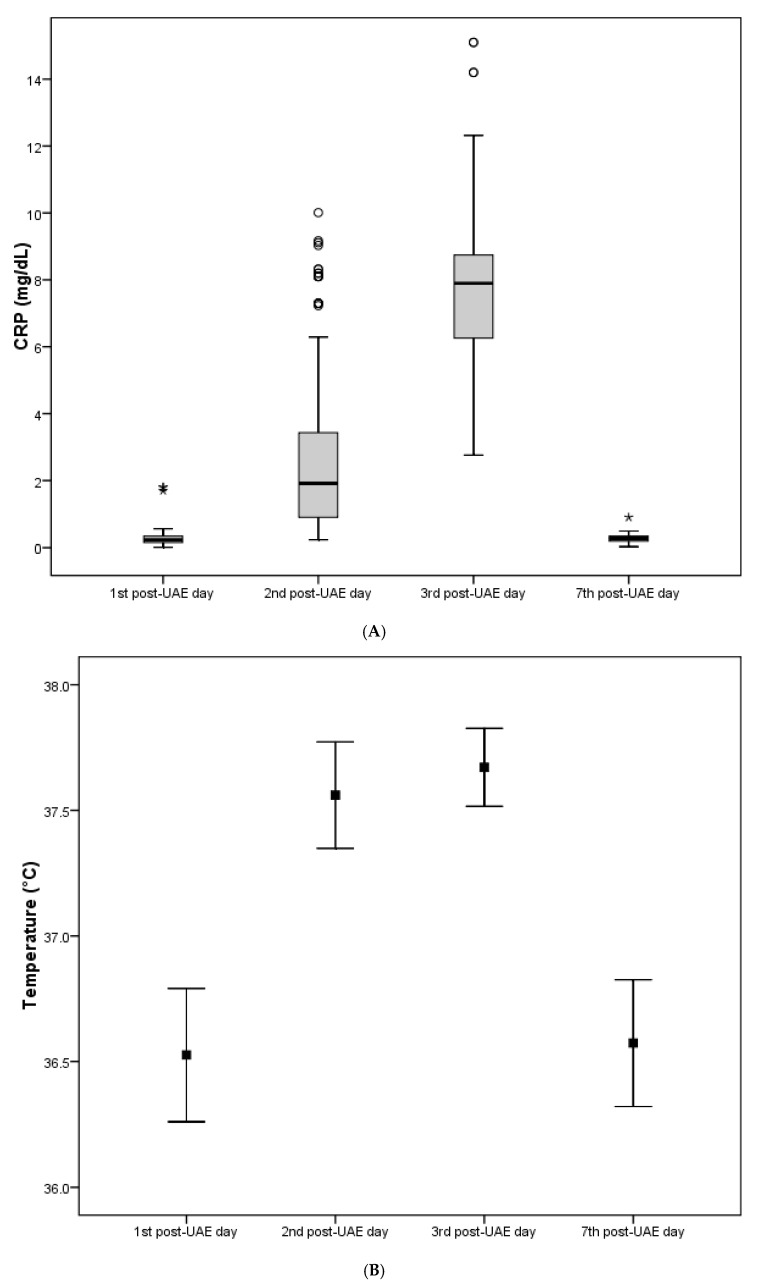
(**A**) The inflammatory parameter CRP among the study participants throughout the follow-up time. (**B**) The inflammatory parameter temperature among the study participants throughout the follow-up time. (**C**) The inflammatory parameter WBC (white blood count) among the study participants throughout the follow-up time.

**Table 1 jpm-13-01385-t001:** A. pre- and post-UAE values of CRP, expressed as the median value and interquartile range (25th to 75th percentile), in relation to hydrogel particles. B. Pre- and post-UAE values of temperature, expressed as mean ± standard deviation (SD), in relation to hydrogel particles. C. Pre- and post-UAE values of WBC, expressed as mean ± standard deviation (SD), in relation to hydrogel particles.

A					
	CRP 1	CRP 2	CRP 3	CRP 4	*p* Value
Total sample	0.23 (0.15–0.34)	1.92 (0.90–3.43)	7.90 (6.26–8.76)	0.27 (0.19–0.34)	<0.001
Particles					
500–700 μm	0.35 (0.20–0.42)	1.00 (0.58–3.99)	6.99 (4.76–8.88)	0.22 (0.14–0.33)	<0.001
700 μm	0.23 (0.15–0.33)	1.96 (0.91–3.43)	7.94 (6.31–8.73)	0.27 (0.19–0.34)	<0.001
700–900 μm	0.23 (0.08–0.31)	0.79 (0.47–3.12)	7.74 (5.99–8.89)	0.28 (0.19–0.39)	<0.001
*p* value	0.083	0.071	0.697	0.537	
B					
	Temperature 1	Temperature 2	Temperature 3	Temperature 4	*p* Value
Total sample	36.532 ± 0.26	37.56 ± 0.21	37.67 ± 0.15	36.57 ± 0.25	<0.001
Particles					
500–700 μm	36.47 ± 0.24	37.52 ± 0.19	37.68 ± 0.12	36.58 ± 0.26	<0.001
700 μm	36.52 ± 0.26	37.56 ± 0.21	37.67 ± 0.16	36.57 ± 0.25	<0.001
700–900 μm	36.57 ± 0.28	37.58 ± 0.20	37.66 ± 0.14	36.56 ± 0.24	<0.001
*p* value	0.606	0.662	0.913	0.983	
C					
	WBC 1	WBC 2	WBC 3	WBC 4	*p* Value
Total sample	7.43 ± 1.48	12.51 ± 1.79	17.67 ± 2.42	6.76 ± 1.50	<0.001
Particles					
500–700 μm	7.10 ± 1.32	13.19 ± 1.08	17.31 ± 2.08	6.87 ± 1.67	<0.001
700 μm	7.46 ± 1.49	12.48 ± 1.84	17.72 ± 2.38	6.79 ± 1.47	<0.001
700–900 μm	7.29 ± 1.45	12.28 ± 1.31	17.30 ± 3.48	6.24 ± 1.78	<0.001
*p* value	0.607	0.262	0.680	0.393	

**Table 2 jpm-13-01385-t002:** Association of pre-UAE values of uterus dimensions and fibroid volume with the maximum values of CRP, temperature and WBC.

	Uterus Dimensions (Pre-UAE)		Fibroid Volume (Pre-UAE)	
	Correlation Coefficient	*p* Value	Correlation Coefficient	*p* Value
CRP	0.069	0.258	0.035	0.567
Temperature	−0.067	0.267	0.071	0.240
WBC	−0.011	0.858	0.141	0.020

**Table 3 jpm-13-01385-t003:** Association between molecular particle size of the embolization agent and the maximum value of CRP, temperature, WBC throughout the follow-up time.

	CRP	Temperature	WBC
Molecular Size			
500–700 μm (16 participants)	7.37 ± 2.74	37.71 ± 0.11	17.32 ± 2.08
700 μm (240 participants)	7.60 ± 1.86	37.72 ± 0.14	17.72 ± 2.38
700–900 (14 participants)	7.31 ± 1.89	37.73 ± 0.14	17.30 ± 3.48
*p* value	0.779	0.941	0.680

## Data Availability

Data of study participants are saved in the Department Obstetrics and Gynecology of Democritus University of Thrace, Greece.

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
