# Peer review of "Our Experience with Hydrogel-Coated Trisacryl Microspheres in Uterine Artery Embolization for the Treatment of Symptomatic Uterine Fibroids and Adenomyosis: A Follow-Up of 11 Years"

_jpm, 2023, doi:10.3390/jpm13091385_

Round 1
Reviewer 1 Report
Dear Authors, description is clear as also many detailed method. It is important to know, that some good results requieres a correct indication of procedure and particles, well done!
Best regards
The reviewer
Author Response
Dear Mr/ Mrs Reviewer
Thank you very much for your encouraging comments. We have also made some additional corrections that we believe further enhance our study.
Reviewer 2 Report
The first paragraph has been repeated in the results sections.
English grammar should be improved.
Spelling errors should be corrected.
English grammar should be improved.
Author Response
The repeated text has been revised in this corrected version.
Some grammatical and spelling corrections have been made in this revised version. I hope the corrected version meets the required quality standards.
Reviewer 3 Report
The study provides valuable information on the use of hydrogel-coated trisacryl microspheres in the UAE for the treatment of symptomatic uterine fibroids and adenomyosis.
Please address the following shortcomings:
1. Lack of a control group: The study does not include a control group, which makes it difficult to compare the effectiveness of the treatment to other methods or a placebo.
2. Limited sample size: The study included 270 premenopausal women aged 33-55 years, which may not represent the entire population suffering from symptomatic uterine fibroids and adenomyosis.
3. Exclusion criteria: The study excluded post-menopausal women, patients with serious comorbidities, patients wishing to preserve their fertility, patients with known allergies to the contrast agent, and patients with suspected malignant conditions. This limits the generalizability of the results to a broader population.
4. Potential complications: The article mentions that the most serious complication is myoma or uterus secondary infection. Treatment failure was reported in cases with previous pelvic surgery, and the percentage of women needing a hysterectomy is 4-30%, especially in cases with less than 10% shrinkage postintervention.
5. Limited follow-up period: Although the study has an 11-year follow-up, it may not be sufficient to evaluate the long-term effectiveness and safety of the treatment.
6. No randomized controlled trials: The study does not mention prospective randomized trials to evaluate the safety and effectiveness of UAE using hydrogel-coated trisacryl microspheres.
7. Variability in treatment outcomes: The reduction in fibroid size and uterus size after UAE can vary, with fibroids shrinking by 40-70% and the uterus size reducing by 40-60%. The effectiveness of UAE in reducing bleeding and pressure symptoms also varies, with bleeding reduction ranging from 80-100% and pressure symptom reduction ranging from 40-60%.
8. Dependence on fibroid characteristics: The reduction in fibroid size depends on the degree of their degeneration, with fibroids having good vascularity expected to shrink more than degenerated ones.
Minor edits
Author Response
- Actually, the study does not include a control group. However, from “historical” data, it is obvious that our method has similar therapeutic results compared to other surgical and/or medical treatments
- The wide spectrum of premenopausal ages “guaranties” that our population represents a satisfactory percentage of the entire population.
- The inclusion of patients with a) serious comorbidities or b) patients with known allergies to the contrast agent could be dangerous or even fatal. Thus, at least, a) and b) are standard criterions of exclusion in most studies and in clinical practice.
…
- The conclusions of many trials, including benign conditions or even cancers are derived after a follow-up of 10 years or even shorter.
- The study does not mention such trials because the main question of the study was the feasibility of the method and its effectiveness in a Greek population and not the comparison with prospective data of trials with a different main purpose in the initial design.
Round 2
Reviewer 2 Report
The first sentence has been repeated in the results section.
Thanks the authors for the corrections.
The English language needs minor editing.
Author Response

(The authors gave the same response as above.)
